# Effects of Straw Mulching on Soil Properties and Enzyme Activities of *Camellia oleifera*–*Cassia* Intercropping Agroforestry Systems

**DOI:** 10.3390/plants12173046

**Published:** 2023-08-24

**Authors:** Huizhen Duanyuan, Ting Zhou, Zhe He, Yuanying Peng, Junjie Lei, Jieyu Dong, Xiaohong Wu, Jun Wang, Wende Yan

**Affiliations:** 1National Engineering Laboratory for Applied Technology in Forestry and Ecology in South China, Central South University of Forestry and Technology, Changsha 410004, China; dyhz990110@163.com (H.D.); hezhe011006@gmail.com (Z.H.); djy4980041@163.com (J.D.); wxh16403@163.com (X.W.); 2China International Engineering Consulting Corporation, Ecological Technical Research Institute, Beijing 100085, China; zt@ciecc.com.cn; 3College of Arts and Sciences, Lewis University, Romeoville, IL 60446, USA; pengyu@lewisu.edu; 4Key Laboratory of Subtropical Forest Ecology of Hunan Province, Changsha 410004, China

**Keywords:** *Camellia oleifera*, intercropping, straw mulching, soil property, soil enzyme activity

## Abstract

In order to explore the influences of rice straw mulching on soil fertility in agroforestry systems, the soil C and N contents and enzyme activities were investigated in a *C. oleifera*–*cassia* intercropping ecosystem in Central Southern China. Three straw mulching application treatments were set up in this study, in 2021, namely, straw powder mulching (SPM), straw segment mulching (SSM), and non-straw mulching as the control (CK). Soil samples were collected from three soil depths (0–10 cm,10–20 cm, and 20–40 cm) in each treatment on the 90th-day after the treatments. The soil organic carbon (SOC), total nitrogen (TN), microbial carbon (MBC), soil enzyme activities (including acid phosphatase (ACP), urease (UE), cellulase (CL), and peroxidase (POD)), and soil water content (SWC) were determined. The results showed that the SOC significantly increased due to the mulching application in SPM and SSM, in the topsoil of 0–10 cm when compared to the CK. The SWC, SOC, TN, and MBC increased by 0.8 and 56.5, 3.5 and 37.5, 21.3 and 61.6, and 5.8% and 76.8% in the SPM and SSM treatments compared to the CK, respectively. The soil enzyme activities of ACP, UE, CE, and POD increased significantly due to straw mulching compared with CK throughout all soil layers. The soil enzyme activities of CL and POD were significantly higher in SSM than in SPM across the soil depth except for ACP. The enzyme activities of ACP were 14,190, 12,732, and 6490 U/g in SSM, SPM, and control, respectively. This indicated that mulching application enhanced the enzyme activity of ACP. Mulching had no significant effects on UE and CL, while the POD decreased significantly in the order of SPM > SSM > CK across all soil layers, being, on average, 6.64% and 3.14% higher in SSM and SPM, respectively, compared to the CK plots. The SOC and MBC were the key nutrient factors affecting the soil enzyme activities at the study site. The results from this study provided Important scientific insights for improving soil physicochemical properties during the management of the *C. oleifera* intercropping system and for the development of the *C. oleifera* industry.

## 1. Introduction

*Camellia oleifera* (tea tree) is a unique woody edible oil tree species in China and is one of the four major woody oil plants in the world, along with olives, oil palm, and coconut, with a history of cultivation and utilization of more than 2300 years [1]. *Camellia oleifera* has considerable social, economic, and ecological value. By 2022, the total planting area of *C. oleifera* plantations reached about 0.7 million hectares in China, and the development of the *C. oleifera* industry and the sustainable management of *C. oleifera* plantations have become the priority goals of the reform of the national edible oil industry [2].

In the *C. oleifera* plantations, the *C. oleifera* seedlings are often intercropped with other crops within a few years after planting in order to utilize and conserve land resources due to the slow expansion of the *C. oleifera* canopy [3]. Therefore, intercropping crop systems have been considered and widely adopted as a traditional management mode in *C. oleifera*–producing areas. However, most crops used in intercropping are annual species, and large amounts of soil nutrients are re-shifted with the harvest of autumn annual crops, which often leads to a degradation in soil fertility in the intercropping systems [4]. Chemical fertilizer is always applied in intercropping systems. However, the application of fertilizer not only incurs financial costs but also results in environmental pollution in *C. oleifera* plantations [5].

Crop straw is one of the main products in agricultural production but is sometimes referred to as the main agricultural waste because these crop straws are often burned by the farmers [6]. The crop straw is rich in nutrients such as nitrogen (N), phosphorus (P), and potassium (K), which is an important factor in maintaining soil fertility. Land cover with straw mulching in intercropping systems is increasingly accepted as an environmentally friendly and low-cost pathway for the utilization of the crop straws [7]. Straw mulching has been identified as an effective way of preventing soil erosion and improving soil quality [8]. Straw mulching can also modify the soil physicochemical properties and increase crop yields in agroecosystems [9,10]. Previous studies have shown that straw mulching can significantly improve the soil water holding capacity, promote crop production, maintain the balance of soil organic matter, and facilitate soil nutrient circulation [4,11]. In addition, straw mulching can reduce annual topsoil loss, improve the efficiency of nutrient supply, influence soil water infiltration and N distribution, and increase the temperature and water content of soils [9,12,13]. The results of an experimental study reported that straw mulching improved the activities of soil amylase, sucrase, cellulase, and peroxidase in the Long Zhong Loess Plateau of China [14]. An experiment with three different crop straws found that covering ginkgo forests with corn straw and rice straw could increase the activities of soil urease, alkaline phosphatase, and sucrase [15]. The straw mulching method has also been applied to *C. oleifera* plantations, but most studies have focused on the effects of straw mulching on the root growth, fruit traits, growth traits, and yields of *C. oleifera* plantations [16]. The effects of straw mulching on the soil nutrient status and enzyme activities in *C. oleifera* intercropping systems is still poorly understood.

In this study, the influence of rice straw mulching on the soil physicochemical properties and enzyme activities was examined in *C. oleifera*–*cassia* intercropping systems. The purpose of this study was to explore the application of mulching to improve soil nutrients and enzyme activities in a *C. oleifera* intercropping system. The objectives of this study were (1) to investigate the changes in soil water content (SWC), soil organic carbon (SOC), total nitrogen (TN), and microbial carbon content (MBC) under different straw mulching treatments and (2) to explore the variations in soil acid phosphatase (ACP), urease (UE), cellulase (CL), and peroxidase (POD) enzyme activities within the soil profile under different straw mulching treatments.

## 2. Results

### 2.1. Changes in Soil Physicochemical Properties

The soil water content (SWC) exhibited a significant increase in the mulching plots when compared to the CK, with the ranking of SSM > SPM > CK particularly observable in the 0–10 cm soil layer (*p* < 0.05). The SWC within the 0–10 cm soil layer was 13.08% higher in SSM and 5.48% higher in SPM compared to that in the CK (Figure 1a). However, the SPM and SSM treatments did not yield statistically significant effects on the SWC within the 10–20 cm and 20–40 cm soil layers (*p* > 0.05) (Figure 1a).

The soil organic carbon (SOC) content exhibited a significant increase in the SPM and SSM plots compared to the CK spanning both the 0–10 cm and 10–20 cm soil layers (*p* < 0.05) and following the ranking SSM > SPM > CK (Figure 1b). Within the 0–10 cm and 10–20 cm layers, the SOC content was significantly higher in SSM compared to SPM (*p* < 0.05) (Figure 1b). The topsoil layer in all straw mulching treatments displayed the highest SOC content, with increases of 41.82% and 28.16% in the SPM and SSM plots, respectively, compared to the CK plots (Figure 1b). As the soil depth increased, the overall SOC contents decreased with a pattern of SSM > SPM > CK (Figure 1b). Within the 0–10 cm soil layer, the soil total nitrogen (TN) content did not exhibit significant differences among the treatments (*p* > 0.05) (Figure 1c), but the straw mulching treatments had significant effects on the soil TN in the 10–20 cm soil layer (Figure 1c). The soil TN ranked as SPM = SSM > CK with an average value of 0.74 g/kg in both the SPM and SSM treatments, representing an ~16% increase compared to the CK plots (Figure 1c). The straw mulching treatments had no significant effects on the soil MBC throughout all soil layers (*p* > 0.05) (Figure 1d).

### 2.2. Changes in Soil Enzyme Activities

The enzyme activities are presented in Figure 2. The soil acid phosphatase (ACP) activities were significantly higher in SPM and SSM plots than in CK plots across all soil layers (*p* < 0.05), with a ranking of SPM = SSM > CK (Figure 2a). The soil ACP was significantly higher in the surface layer (0–10 cm soil depth) than in the deeper 10–20 cm and 20–40 cm soil layers (*p* < 0.05) (Figure 2a).

The soil urease (UE) activities were higher in both SPM and SSM treatments than in the CK within the 0–10 cm soil layer, but no significant effect was observed with changing soil depth (*p* > 0.05) (Figure 2b). The soil UE activities of both SPM and SSM were 51.0% and 49.3% higher than that of CK, respectively (Figure 2b).

The SPM and SSM treatments had no significant effects on the soil CL except for the 20–40 cm soil layer (*p* > 0.05) (Figure 2c). The soil depth had no significant effect on the soil CL as well (Figure 2c).

The soil peroxidase (POD) activities exhibited significant increases in both SPM and SSM treatments when compared to the CK treatment, and the ranking was SSM > SPM > CK within the 0–10 cm and 10–20 cm soil layers (*p* < 0.05) (Figure 2d). The soil POD activities were 44.3% and 12.4% higher in the SPM and SSM plots than in the CK plots (Figure 2d).

### 2.3. Correlation between Soil Nutrients and Enzyme Activities

The Pearson’s correlation analysis revealed significant positive correlations between soil nutrients and enzyme activities (*p* < 0.05, Table 1). Among them, a strong positive correlation with a coefficient of 0.836 was observed between SOC and MBC. Significant positive correlations were also identified between SOC and TN, as well as SOC and S-UE with correlation coefficients of 0.754 and 0.766, respectively (*p* < 0.05, Table 1). There was a significant positive correlation between TN and MBC with a correlation coefficient of 0.815 (*p* < 0.01, Table 1). In addition, significant positive correlations were found between the soil MBC and both S-UE and S-ACP with correlation coefficients of 0.711 and 0.668, respectively (*p* < 0.05, Table 1); moreover, a significant positive correlation existed between S-UE and S-ACP with a correlation coefficient of 0.772 (*p* < 0.05, Table 1).

## 3. Materials and Methods

### 3.1. Experimental Location

The experimental area was located in the Baisha Village, Changsha County, Hunan Province, China (111°53′–114°15′ E and 27°51′–28°41′ N) (Figure 3). This area belongs to the subtropical monsoon climate zone, with a mild climate and four distinct seasons. The annual average temperature was 16.8–17.3 °C; the annual accumulated temperature was 54–57 °C; and the annual maximum and minimum precipitations were 1961 mm and 934 mm, respectively. The experiment was conducted in a young, three-year *C. oleifera* plantation with a planting distance of 3 × 3 m in March of 2021.

### 3.2. Study Materials and Experimental Design

In this study, the *C. oleifera*-growing areas were selected as experimental site and used agricultural rice straw for mulching treatment. The straw mulch contains an SOC of 36.0 g/kg, a TN of 11.4 g/kg, and a TP of 1.1 g/kg (Figure 4a, CK). The straw was naturally air-dried and then processed into a powder (Figure 4b, SPM) and 5–7 cm segments (Figure 4c, SSM) for the mulching treatments.

In this study, the *cassia* seeds were placed in the experimental area in March 2021 with distances between each hole of 0.3 m. The distance between the *cassia* holes and the *C. oleifera* trees in this intercropping system was 0.5 m. Mature *C. oleifera* and *cassia* plants were harvested in September 2021. After harvesting, the experimental area was divided into a total of 15 study plots each with an area of 3 m × 7 m (21 m^2^), and three *C. oleifera* trees were left in each plot. A trench was dug between the boundaries of each study plot. A corrosion-resistant, environmentally friendly plastic partition with a dimension of 7 m × 0.6 m × 3 cm (length × width × height) was placed in each trench, to separate the boundaries of the study plot and prevent interactions between the different sections, and then filled with soil.

The local rice straw was collected, naturally air-dried, and then milled into straw powder using an agricultural grinder with straw segments of sizes ranging from 5 to 7 cm.

A split-plot design was carried out in this study. The three types of treatments including straw powder mulching (SPM), straw segment mulching (SSM), and non-straw mulching (control treatment, CK) as the main factor were set up in the study site. Each treatment involved a 7 m × 3 m (21 m^2^) sample plot replicated five times, resulting in a total of 15 plots for this study. The three soil depths (0–10, 10–20, and 20–40 cm) as the sub-factor were nested within each plot. A total of 45 (15 × 3) soil samples were collected for soil property measurements. To collect these soil samples, five soil sampling points (four directions and one central point) were established in an X-shaped distribution within each plot, maintaining consistent elevation and slopes. The soil samples were collected from the three soil depths using a soil auger with an inner diameter of 4.5 cm. Samples from the same soil depth within each plot were pooled as one mixed soil sample. The straw was applied to cover an area of 1 square meter around the base of each *C. oleifera* stem. The quantity of straw used for mulching around each *C. oleifera* tree was 0.8 kg/m^2^, equivalent to the annual yields of rice straw per square meter in the study area.

### 3.3. Collection and Soil Analyses

Rice straw mulching was initiated on 9 October 2021, and the soil samples were collected on the 90th day after treatment application. In brief, debris such as dead leaves and straw on the soil surface were removed. Then, a ring knife with a volume of 100 cm^3^ and a soil drill with an inner diameter of 5 cm were used to extract soil samples from three soil depths around the base of the *C. oleifera* stems in each plot to determine the SWC. Stones and roots were singled out from all soil samples, followed by sieving of the soil through a 0.15 mm sieve. The sieved soil was then stored in a refrigerator freezer at 4 °C to determine the soil enzyme activities. The soil water content (SWC) was assessed using the 105 °C drying method. The soil organic carbon (SOC) was quantified using the oxidation method with K_2_Cr_2_O_7_–H_2_SO_4_ followed by titration with FeSO_4_ [17]. The total nitrogen (TN) was measured using the Kjeldahl method employing the HGK–99 automatic Kai nitrogen-fixing instrument (Shanghai Heguan Company, Shanghai, China). The microbial biomass carbon (MBC) was initially fumigated with CHCL_3_ and then extracted using K_2_SO_4_. The procedures for evaluating the soil physicochemical properties adhered to the guidelines outlined in the third edition of *Soil Agrochemical Analysis* by Bao [18]. The activities of each soil enzyme were analyzed in accordance with the instructions of the soil enzyme assay kit (Beijing Solebao Biotechnology Co., Ltd., Beijing, China) [19].

### 3.4. Statistical Analysis

Statistical tests were performed to determine the effects of the three treatments (SPM, SSM, and control) and the soil depth on soil physicochemical properties and enzyme activities by using two-way analysis of variance (ANOVA). The significant difference between the treatments was tested using the Duncan method (*p* < 0.05). Correlation analysis was used to find out whether there was a relationship between the soil nutrient content and the soil enzyme activity variables, and then, the magnitude and effect of this relationship in the *C. oleifera*–*cassia* intercropping system was studied. The Pearson’s correlation coefficient was a measure of linear association. The original data were log-transformed to satisfy the normality and homoscedasticity assumptions of ANOVA. The statistical analyses were conducted using the SPSS (22.0) or SAS statistical package [20].

## 4. Discussion

The rice straw mulching treatments exhibited effective enhancement of the soil water content (SWC) within the *C. oleifera*–*cassia* intercropping system in our study (Figure 1). It was also observed that straw mulching led to increased soil nutrients levels and improved activities of ACP, UE, CL, and POD (Figure 2).

The soil water content (SWC) stands out as one of the most vital physical indicators of soils, and it is also an important indicator of soil actions and effects affecting plants. The SWC is an essential factor for controlling plant growth and development [21]. In the *C. oleifera*–*cassia* intercropping system, rice straw mulching played a crucial role in improving the SWC in our study. This effect is mainly attributed to the reduction in soil temperature variations, as well as the reduction in soil evaporation and transpiration through the mulching treatments [22]. Our research was consistent with previous investigations such as those demonstrating substantial impacts of straw mulching on the SWC within the 0–20 cm soil layer subsequent to the implementation of straw coverage [23].

The application of mulching treatments in our study led to an increase in the soil organic carbon (SOC) content; this trend is consistent with the findings of another experiment. This previous study reported an increase in SOC within the 0–20 cm soil layer after straw coverage, indicating a rise from 1.2 g/kg in the control treatment (CK) to 6.1 g/kg in the mulching treatment [24]. Similarly, the SOC content increased by 50–58% as a result of the mulching treatments in our study. The results can be attributed to the accumulation of additional nutrients due to straw mulching. This accumulation might contribute to improved soil structural stability, soil fertility, and plant productivity [25]. Maintaining an optimal level of SOC in soil by straw mulching further enhances the filtration capacity of soils, which in turn contributes to providing clean water [26]. The straw mulching treatment enhanced the photothermal properties of the soil, increased soil microbial activity, and promoted the accumulation of SOC [27]. In this study, the SOC demonstrated a decline with increasing soil depth. This pattern could be attributed to a decrease in the organic material input within deeper soil layers, leading to a reduction in both internal and external influences [28]. The surface soil exhibited a higher SOC content compared to the deeper soil layers, and the application of straw mulching treatments resulted in increased SOC levels compared to the control in our study. These outcomes can be attributed to the significant contribution of straw mulching on the soil surface as a crucial source of soil organic matter. This allows the conversion of soil organic matter to inorganic nutrients, enhancing both the quantity and activity of soil microorganisms in the surface soil [29]. The SOC content was found to be higher in the SSM treatment than in the SPM treatment in our study. This difference might be attributed to the increased accumulation of substances and reduction of water and wind erosion by SSM treatment [30]. In addition, the SSM treatment had a stronger effect on improving soil insulation and supporting microorganism growth in the soil. This led to increased microbial activity, subsequently resulting in the secretion of more soil enzymes that enhance soil biochemical activities, promoting the release of greater amounts of soil nutrients [31]. The presence of mulch material reduces soil erosion directly, while also indirectly enhancing the decomposition of soil organic matter [32].

Nitrogen is an essential nutrient for plant growth (structure), plant food processing (metabolism), and the creation of chlorophyll. Soil nitrogen is mainly decomposed, transformed, and preserved in organic matter [33]. Straw mulching can increase the soil mineral N and the N-uptake efficiency of wheat plants [9]. Straw mulching also generates a favorable habitat for the soil microbial communities’ composition, changes the N in the soil via changes in soil N mineralization, and, consequently, increases the soil mineral N and the total N availability [34,35,36]. However, no significant effects were observed in response to the mulching treatments in this study. This outcome could potentially be attributed to the N saturation in forest soils [37]. This observation is consistent with the findings of other researchers who have also reported the lack of a distinct relationship between soil fertility and straw mulching [29].

The soil MBC content indicated an increase due to mulching, with the most significant impact observed in the surface soil in this study. Our results agreed with the findings that wheat straw and corn straw mulching treatments can sustain elevated MBC levels by promoting an increase in the soil microbial populations and their activities [38].

The soil enzyme activities serve as a crucial indicator of soil fertility, soil quality, and soil health [39]. Soil enzyme activities arise from a combination of crop root secretion and the stimulation of soil microorganisms, playing an important role in the decomposition of organic matter and the nutrient cycling [40]. Following straw mulching, the activities of all four soil enzymes exhibited improvement. The correlation analysis additionally highlighted a significant positive relationship between soil carbon nutrient content and soil enzyme activities in this study. The contents of organic substances including organic carbon in the soil under straw mulching promoted the transformation of organic matter to inorganic forms. This process provided a substantial carbon source for the activities of bacteria, fungi, and other soil microorganisms. As a result, the soil enzyme activity was enhanced, contributing to an overall improvement in soil health [41].

The soil UE is the sole enzyme responsible for catalyzing urea hydrolysis. Our results demonstrated a significant enhancement in UE activities due to straw mulching. This observation was in line with results suggesting that UE activity is highest within the 0–10 cm soil layer and that straw mulching has a significant impact on enzyme activities within this depth range [42]. The UE activities exhibited a significant and positive correlation with the SOC. Previous studies have also indicated that UE activities can be influenced by the soil carbon source [43].

The present study revealed that ACP activities were higher in the straw mulching treatments than in the CK across all soil layers. This observation is supported by previous findings indicating that the ACP in the soil beneath straw cover was significantly greater than that in the CK across various materials used for vineyard coverage. In addition, the straw cover treatment demonstrated an improvement in ACP activities, displaying a decreasing pattern with an increase in soil depth [44]. Our results were also consistent with the findings that straw mulching could supply the carbon source of the soil and improve the activity of CL [45]. The soil CL activities were positively correlated with the SOC in this study, which was in agreement with the findings that the POD enzymes were directly involved in the soil carbon cycle, the mineral element cycle, and energy flow in soil ecosystems, and they played an important role in the biogeochemical cycles of soil C and N [46]. The enzyme activities of POD were found to be higher in mulching than in CK, further supporting the association between elevated POD activities and a higher organic matter content within the soil [47]. The observed POD activities were associated with the soil carbon cycle, leading to an increase in the transformation of soil organic carbon and a subsequent improvement in soil quality. The SSM treatment exhibited superior effects compared to the SPM treatment on SOC, TN, SWC, MBC, and enzyme activities in the surface soil in this study. The SSM treatment showed better results compared to the SPM across various parameters, except for CL activity [48]. The SSM offers a greater abundance of substrates for soil enzymes and their activities. On the other hand, straw mulching maximizes the soil water content and the organic substrate in the topsoil. This increased availability of organic matter coupled with increased soil moisture encourages the growth of microorganisms, and the favorable environment conditions enhanced the accumulation of enzyme activities [49].

## 5. Conclusions

The application of rice straw mulching led to a significant increase in the soil nutrients, SWC, MBC, and microbial activities within the *C. oleifera*–*cassia* intercropping system, as compared to the control treatment in this study. The SSM method exhibited more pronounced effects on soil physical, chemical, and biological properties compared to the SPM treatment. In addition, a positive correlation was observed between soil enzyme activities and soil organic carbon (SOC) as well as soil MBC. The implementation of rice straw mulching holds the potential to address challenges related to the soil fertility reduction resulting from crop harvesting in the *C. oleifera*–*cassia* intercropping system. This study provided a strong scientific basis and technical guidance for the sustainable management of *C. oleifera* plantations, contributing to the maintenance of soil fertility and overall soil health in *C. oleifera*–crop intercropping systems.

## Figures and Tables

**Figure 1 plants-12-03046-f001:**
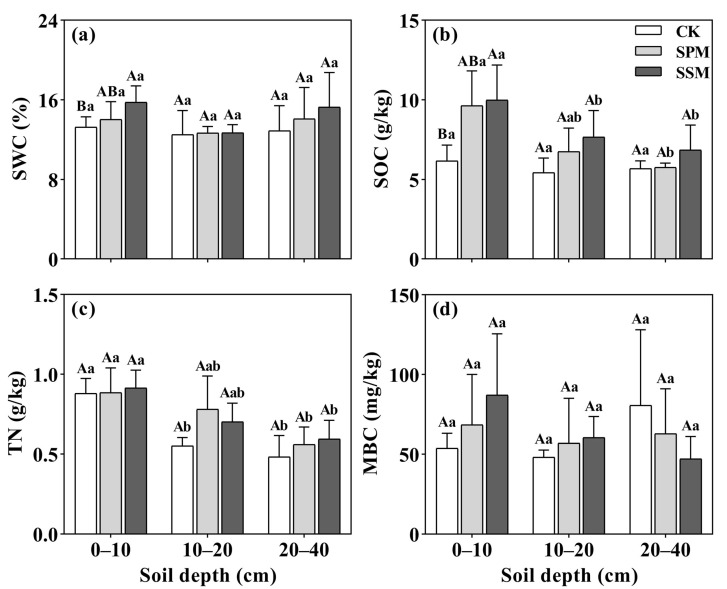
Changes in physical and chemical properties of the soil under three straw mulching treatments in the *C. oleifera*–*cassia* intercropping system: SWC (**a**), SOC (**b**), TN (**c**), and MBC (**d**). SPM: straw powder mulching treatment, SSM: straw segment mulching treatment, and CK: non-straw mulching. Different capital letters mean the difference between the same soil layers of different treatments, and the lowercase letter means the difference between different soil layers with the same treatment. The different letters indicate the significant differences (*p* < 0.05).

**Figure 2 plants-12-03046-f002:**
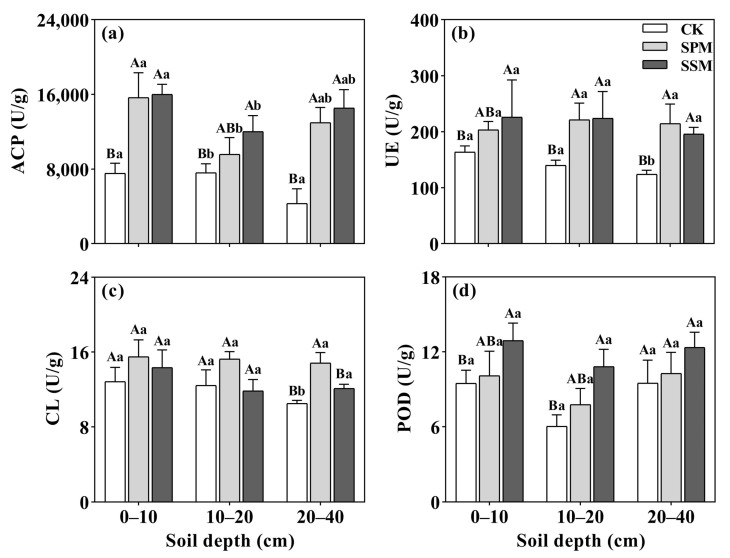
Changes in soil enzyme activities under three straw mulching treatments in the *C. oleifera*–*cassia* intercropping system: ACP (**a**), UE (**b**), CL (**c**), and POD (**d**). SPM: straw powder mulching treatment, SSM: straw segment mulching treatment, and CK: non-straw mulching. Different capital letters represent the difference between the same soil layers of different treatments, and the lowercase letter represents the difference between different soil layers with the same treatment. The different letters indicate the significant differences (*p* < 0.05).

**Figure 3 plants-12-03046-f003:**
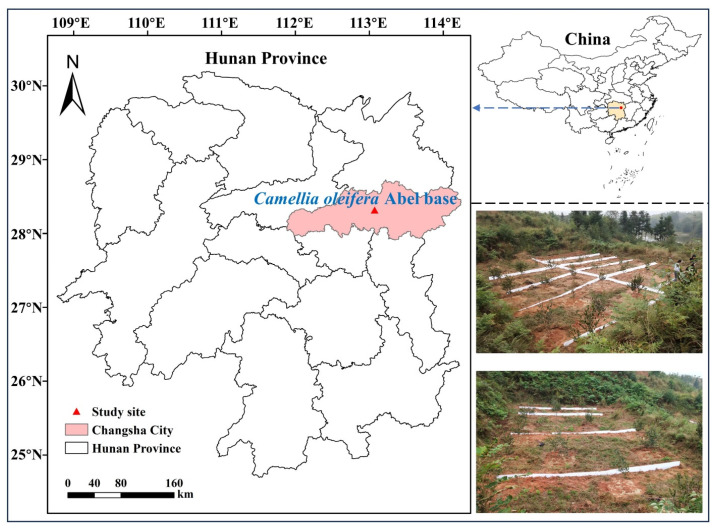
Experimental location and distribution.

**Figure 4 plants-12-03046-f004:**
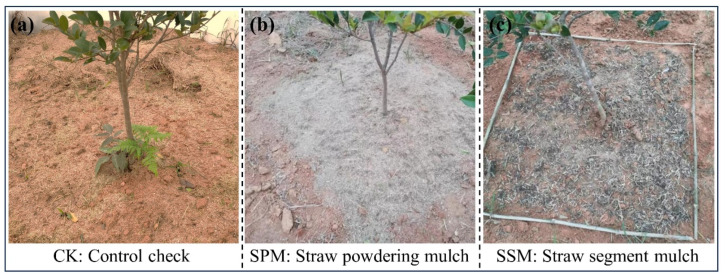
Three experimental treatments of this study: control (**a**) CK, straw powder mulching (**b**) SPM, and straw segment mulching (**c**) SSM.

**Table 1 plants-12-03046-t001:** Correlation coefficient of soil nutrient content and soil enzyme activity in the studied *C. oleifera*–*cassia* intercropping system.

	SOC	TN	MBC	S-UE	S-ACP	S-CL	S-POD
SOC	1						
TN	0.754 *	1					
MBC	0.836 **	0.815 **	1				
S-UE	0.766 *	0.462	0.711 *	1			
S-ACP	0.57	0.489	0.668 *	0.772 *	1		
S-CL	0.46	0.597	0.65	0.565	0.657	1	
S-POD	0.578	0.259	0.474	0.664	0.467	−0.01	1

Note: * Represents a significant correlation with (*p* < 0.05), and ** represents a significant correlation with (*p* < 0.01).

## Data Availability

The data presented in this study are available on request from the corresponding author. The data are not publicly available because the funded projects have not been completed.

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
