# Peer review of "Effects of Straw Mulching on Soil Properties and Enzyme Activities of Camellia oleifera–Cassia Intercropping Agroforestry Systems"

_plants, 2023, doi:10.3390/plants12173046_

Round 1
Reviewer 1 Report (Previous Reviewer 1)
Dear Authors,
in this new submission the manuscript was improved, but the main previous problems were not solved.
My most serious criticism is the same as last time. The statistical analysis of the data is not fully explained. In fact, Table 1 reports a "correlation" of which there is no trace even in 2.4 Statistical analysis. It is mandatory in a scientific article for Plants. All statistical analyses must be explained in sufficient detail to replicate the methodology.
The review of the style and language is mandatory at this point, since I still found the same errors reported previously.
The observations are in this list, but please consider the attached file for better verification.
Line 22 and 24 Please, check (Soil water content (WC)
Line 89-91 please check the sentence.
Line 125-127 Please check the sentence.
Lines 130 Please check the sentence. (WAS?)
Lines 142 (from the roots of C. Oleifera) Please, explain.
Line 144 and 145,(0C)…or better °C Please, check in the manuscript.
Line 154 2.4. Statistic Alalysis: check the spelling. Moreover, in this paragraph the correlation analysis is not explained. See table 1 (lines 215 and 225-228)
Line 234-235 Check the sentence.
Line 245-247 These sentences need to be revised.
Line 263-266 Please, explain better. This sentence is not clear.
Line 267-271 These are trivial considerations and do not contribute to the discussion.
Author Response
Please see the attached file

Reviewer 2 Report (Previous Reviewer 3)
The manuscript entitled: "Effects of straw mulching on soil properties and enzyme activities of camellia oleifera-cassia intercropping agroforestry systems" written by Huizhen et al., 2023. The paper has been improved greatly, and the authors have tried to resolve all the concerns. I believe the manuscript has potential to be published in "Plants". However, I still have concerns related to the study applications in the field and its novelty. I have following questions to the authors.
1: The study was conducted in march 2001, harvesting was done in 09-2001. Later on, Rice straw was covered on October 9, 2021, and the soil samples were taken on the 90th-day post-treatments. I am confused between these statements. The methodology should be explained further.
2: Why the data has been presented in 2023 for publication when the study was carried out in 2001 and sampling was even done in 2001? Please explain with reasons. if scientifically, explain in the methods section.
3: Can you give an example in the introduction where possible effects of powder and segment mulching (or anything relevant) has been evaluated in the past?
4: Add "The" in line 28.
5: ANOVA was one-way or two-way?
6: Results are fine.
Dear Editor;
The English quality of this manuscript seems fine. Only a brief revision is required for some typo errors or somewhere grammar mistakes.
Round 2
Reviewer 1 Report (Previous Reviewer 1)
Dear Authors,
mulching is not only an agricultural practice, but used to reduce impacts in forestry operations.
Thank you for taking the proposed comments into consideration. There are still two small changes to be made, only editorial.
They are:
1) capital letter (camellia) in the title
2) add the zero in front of the decimal numbers (.XXX) in table 1
This manuscript is a resubmission of an earlier submission. The following is a list of the peer review reports and author responses from that submission.
Round 1
Reviewer 1 Report
The manuscript deals with an interesting problem concerning the conservation of soil fertility, using residues from processing of other agricultural crops.
However, some essential points for a scientific publication are missing and must be clearly explained.
It is absolutely necessary to indicate the statistical analyzes applied and the conditions of application. Indeed, some results proposed in the figures and commented do not seem congruent.
The authors must add to the manuscript the necessary explanations in methods that make it possible to evaluate the results, the discussion and the conclusions. It is absolutely necessary to indicate the statistical analyzes applied and the conditions of application. Some preliminary notes are added to the manuscript file.
Reviewer 2 Report
The manuscript presents a poor experimental design with few analyses and a poor discussion of them. However, I believe that by modifying some of the parts and taking into account the comments made, it could be published.
General comments:
1. Rewrite the abstract following the recommendations below.
2. Try to be consistent in the verbal tense used during the manuscript.
3. Improve the quality of the figures, some of them are quite blurred.
4. In general, I would revise the English of the whole manuscript.
Detailed comments:
1. Ln19-22. Rewrite this sentence because it is hard to understand. I recommend to rewrite this sentence as follows:
“The treatments SPM and SSM increased some soil parameters: 0.8% and 21.3% in water content (WC), …”
2. Ln22. “SSM was better…”. Better for what? Introduce the results before assuming these conclusions, because the lector has nothing to probe that conclusion. Anyway, it is better to say “SSM, based on our results, seems to be better than the treatment X BECAUSE…”
3. Ln24-28. The same as in the suggestion for Ln19-22.
4. Ln53. The crop straw contain OR is rich, but not both verbs.
5. Ln59. As a suggestion, put the reference just after the citation of the paper, not at the final of the sentence.
6. Ln69. It is not a unique experiment, it is a research work, so I recommend to describe it as “study”.
7. Ln84-90. Were these data obtained the year of the experiment or are data from the annual average?
8. Section 2.1. I would change this title to “Experimental location”. Moreover, I recommend to write the experimental description in past simple tense, because the experiment was already developed.
9. Section 2.1. Could you explain a bit more about the forest under study?
10. Figure 1. Change the title to “Experimental location and distribution.”
11. Section 2.3. As in the comment number 8, I recommend to write this part in past simple tense. Rewrite this part being consistent in the verbal tense used.
12. Ln102-103. “…between experimental land…”? I don’t understand this sentence.
13. Ln114. “Each treatment was randomly repeated for 5 times”. Do you refer to the replicates? Please, specify.
14. Section 2.4. Change the title to “Collection and soil analyses”.
15. Ln125-127. Explain better the composition of each soil sample, it is not clear.
16. Section 2.4. Could you explain in detail the methods used and provide their references?
17. Section 2.5. Change the title to “Statistical analyses”. Explain more in detail the data analyses.
18. Figure 3. Change the tile. Here you must explain what are the graphs explained. I recommend to entitled it as “Soil physical and chemical properties of the soil.” Then, I would describe all the terms used in the graphs and the differences observed with the statistical analyses applied.
19. Section 3.2. “Changes in soil enzyme ACTIVITIES”.
20. Ln209. Delete “basically”.
21. Ln210. Delete “very”.
22. Ln213. Delete “very”.
23. Ln224-225. Rewrite the sentence, I can’t understand it.
24. Discussion part. In general, rewrite it in order to discuss the results and not repeat the results obtained.
25. Ln305-309. Rewrite, hard to understand.
Must be improved. The authors must check the English in the whole manuscript
Reviewer 3 Report
I have reviewed the manuscript written by DuanYuan et al., 2023 and found the study very interesting. I could say that this research has great potential for future applications. The work is organized very well and can be published in the "Plants" after major revisions. I hereby raised some concerns for the authors to satisfy them.
1: The researchers found that the SSM technique was more beneficial than SPM and CK. Do they have any explanation for it? Because the contents of both SPM and SSM are the same, except for the way of application.
2: Can the author include some facts about such kind of mulching patterns in the introduction from the previous literature? I can see some examples of forest soils, however, I was unable to see any cultivated crop data in the introduction section.
3: The methods and results sections are presented very clearly.
4: Conclusion should be re-written based on the obtained results.
Dear Editor;
I am satisfied with the quality of the English language of this manuscript.
Regards
Round 2
Reviewer 1 Report
Dear Authors,
you have improved the manuscript. However, a revision of the style and language is still necessary because it is sometimes particularly difficult to understand the meaning of the speech, especially in the discussion paragraph.
I appreciated the title change and the clarification of the agroforestry system. By the way, in the text of the manuscript you state that it is a forest of Camellia oleifera. From the pictures it is clear that it is a plantation and not a forest. Although built with forest trees, it is actually a plantation (planting distance 3x3). So, I explicitly ask you to remove the indications "forest" and indicate that it is a plantation. If it is a forest, explain the reasons and a better argument is needed.
My most serious criticism however is still related to the statistical analysis of the data, which is not explained thoroughly.
Some observations are in this list, but please consider the attached file for better verification.
Line 327 Please, check
Line 806 Fig 2 the figure is missing
Lines 814_819 (The three-soil depth with layers (0–10, 10–20, and 20–40 cm) as the sub-814 factor in each experimental plot. The total soil sampling points for examinations of soil 815 properties was 45 (15 x 3) in this experiment points which was mixed with a S-shaped. 816 The straw was covered within 1 square meter of the root of C. oleifera, and the C. oleifera in 817 the same plot with the straw of the same form. The amount of straw at the root of each C. 818 oleifera tree was 0.8 kg/m2, which was the local seasonal straw yield per square meter.)
These sentences are not clear. Please rewrite.
Line 1688 (WC)
please check
Lines 1700-1703 (SOC content increased by matching treatments in our study, which is supported by another experiment of two straw mulching treatments, they found that the SOC of 0–20 cm soil layer increased after straw covering, the SOC content of 0–10cm soil reached at 6.1 g/kg in mulching than in CK at 1.2 g/kg, as well as the results)
This sentence is not clear. Please rewrite.
Lines 1703- 1706 (The SOC content increased ~ 50%−58% of soil organic matter , which accumulates more soil nutrients for plants, improves soil structural stability, and to enhance soil fertility and plant productivities [18]. )
This sentence is not clear. Please rewrite.
Lines 1709-1711 (SOC decreased in this study, it should be explained that the deeper the soil layer, the less input of organic substances, and weaken the external and the internal influences in the deep soils)
Please, check the sentence.
Lines 1711-1715 (The straw mulching on soil surface is an important source of soil organic substances, and these resources were great supplements of soil organic matter and SOC, and then to accelerate the conversion of organic substances to the inorganic carbon of soils, as well as the straw mulching treatment increased the number of soil microorganisms in 0–10 cm soil layer)
Please, check the sentence.
Line 1946-1948 (…provide sufficient carbon source for the activities of bacteria, fungi and other microorganisms in the soil, and promotes the improvement of soil enzyme activity [31].)
Is it only a positive effect or can it induce phytopathological problems? What did you observe?
Lines 1962-1965 They are not completed.

Reviewer 3 Report
I have read the revised version of the manuscript written by Huizhen et al., 2023. I found that the manuscript has been extensively improved after revision. I am satisfied with the presentation of results and the statistical analysis for differentiation between data. However, I found some issues with the English quality of this manuscript. For example, the last paragraph of the "Discussion part" has a few mistakes.
Overall, I would suggest acceptance of the study in "Plants".
The English grammar needs to be checked throughout the manuscript.